# Awareness and Factors Associated with Health Care Worker’s Knowledge on Rubella Infection: A Study after the Introduction of Rubella Vaccine in Tanzania

**DOI:** 10.3390/ijerph16101676

**Published:** 2019-05-14

**Authors:** Nikolas A.S. Chotta, Melina Mgongo, Jacqueline G. Uriyo, Sia E. Msuya, Babill Stray-Pedersen, Arne Stray-Pedersen

**Affiliations:** 1Institute of Clinical Medicine, Faculty of Medicine, University of Oslo, 0863 Oslo, Norway; linnabenny@yahoo.com (M.M.); babill.stray-pedersen@medisin.uio.no (B.S.-P.); arne.stray-pedersen@medisin.uio.no (A.S.-P.); 2Better Health for African Mother and Child, P.O. Box 8418, Moshi, Tanzania; jackieuriyo@yahoo.com (J.G.U.); siamsuya@hotmail.com (S.E.M.); 3Department of Community Health, Institute of Public Health, Kilimanjaro Christian Medical University College, P.O. Box 2240, Moshi, Tanzania; 4Department of Epidemiology and Biostatistics, Institute of Public Health, Kilimanjaro Christian Medical University College, P.O. Box 2240, Moshi, Tanzania; 5Department of Forensic Sciences, Oslo University Hospital, 0863 Oslo, Norway

**Keywords:** healthcare workers, immunization, knowledge, rubella, vaccine

## Abstract

*Background* Congenital rubella syndrome is a global health problem. The incidence is much higher in Africa and Southeast Asia than the rest of the world, especially in countries where universal rubella vaccination has not been implemented. Healthcare worker’s knowledge on rubella infection and the rubella vaccine is of utmost importance in achieving and maintaining vaccination coverage targets. This study aimed to assess health care workers knowledge on rubella infection in Kilimanjaro Tanzania, after the introduction of a rubella vaccination. *Methods* This was a health facility-based cross sectional study. It was conducted in three districts of the Kilimanjaro region between August and October 2016. The study involved eligible health care workers in selected health facilities. An interview guide was used for collecting information by face-to-face interviews. Multivariate analysis was used to assess factors associated with rubella knowledge among healthcare workers. *Results* A total of 126 health care workers were interviewed. An acceptable level of knowledge was considered if all five questions about rubella were correctly answered. Only 26.4% (*n* = 31) answered all questions correctly. In multivariate analysis education level and working department were predictors of rubella knowledge; health care workers with an advanced diploma had an adjusted odds ratio (AOR) of 7.7 (95% Confidence interval; CI: 1.4, 41.0), those with a university degree (AOR: 10; 95% CI: 2.4; 42.5) and health care workers in the outpatient department (AOR: 0.06; 95% CI: 0.04; 0.29). *Conclusions* Our study confirmed that health care worker’s knowledge on rubella infection was low in the areas where rubella vaccination had been introduced. We recommend continuous education and supportive supervision post vaccine introduction in order to increase healthcare worker’s knowledge on rubella infection, congenital rubella syndrome and prevention through sustained high vaccination coverage.

## 1. Introduction

Rubella is an acute childhood illness caused by the Togavirus of the genus rubivirus. It presents with maculopapular rashes [1]. Primary rubella infection among women in the first trimester of pregnancy carries high risk (>90%) of transmitting the virus to the fetus [1,2]. Fetal infection may result in fetal death or various malformations in congenital rubella syndrome (CRS). Congenital rubella syndrome is an important cause of severe birth defects; it affects the eye, ear, heart, spleen, liver and brain. Affected children present with mental retardation, microcephaly, developmental delay, impaired vision/blindness, hearing impairment and congenital heart diseases [1,2].

Congenital rubella syndrome is a global health concern. Data shows that the global incidence is 0.1 to 0.2 per 1000 live births and more than 100,000 infants are diagnosed with CRS annually [3]. The incidence may increase by twenty-fold during rubella epidemics. Low incidence is found in areas where the rubella containing vaccine (RCV) has been implemented [4]. In Africa and Southeast Asia, where RCV vaccination was not fully implemented, the incidence of CRS was at its highest with up to 121 per 100,000 live births in the year 2010 [4]. 

According to the Global measles and rubella strategic plan (2012–2020), eliminating measles and rubella requires achieving and maintaining high levels of population vaccination coverage. This can be achieved by engaging the public and other sectors to build confidence and increase the demand for vaccinations. To achieve these objectives health care workers need excellent knowledge and skills for promoting vaccination [5,6,7,8,9,10].

Studies in developed and developing countries have reported knowledge gaps on vaccinations among health care providers [11,12,13,14]. These gaps in vaccine knowledge among health care workers (HCW’s), limits communication and interaction with parents, resulting in poor vaccine uptake [15,16].

The healthcare worker’s knowledge on vaccination is influenced by several factors, including awareness and attitude towards vaccines, beliefs which align with scientific truth, continuous medical education, supportive supervision, personal experience, specialty type, knowledge base and training [11,14,16,17].

In Tanzania, the rubella vaccine was introduced in 2014, the vaccination coverage was above 95% [10]. There have been no studies assessing the health care worker’s knowledge on rubella infection and vaccine. Detailed knowledge on rubella is needed in order to respond to specific vaccination barriers and formulating appropriate techniques to overcome these barriers [16]. Furthermore, the surveillance on rubella, monitoring effectiveness of rubella vaccination, diagnosing congenital rubella syndrome and leading a multi-sectorial approach to sustain high vaccination coverage, requires HCW’s with good knowledge [16].

This study therefore aimed to assess health care worker’s knowledge on rubella infection, post rubella vaccine introduction. The study explores the existing knowledge gaps and provides evidence for planning interventions.

## 2. Materials and Methods

### 2.1. Study Design and Area 

This was a quantitative cross-sectional study, which was conducted from September to October 2016, in three districts out of seven districts of the Kilimanjaro region. The three districts, namely Moshi Municipal Council, Rombo and Same District councils were selected to represent urban and rural settings. Moshi Municipal Council is the capital city of the region and is more urban compared to Same and Rombo districts. The Kilimanjaro region is located in North Eastern Tanzania, it has a high number of health facilities covering the dispensary level to a zonal referral hospital. 

Kilimanjaro region has the highest vaccination coverage in Tanzania at 93% versus the national vaccination coverage of 75% [17].

### 2.2. Study Population 

The study population included health workers directly involved in providing medical services to patients (doctors, nurses, medical technicians and medical attendants), working in the selected public health facilities located in the three districts of the Kilimanjaro region.

### 2.3. Sampling Methods

The sample size was calculated using the sample size calculator from Creative Research Systems: Survey software, assuming 50% of the total 485 HCWs in the selected facilities would have adequate knowledge, with a confidence interval of 10, at a 95% confidence level, the minimum calculated sample size was 120.

A multistage sampling technique was used to select urban and rural representative health facilities. The seven districts in the Kilimanjaro region were categorized into three urban and four predominantly rural districts. The districts were written in equal sized pieces of paper, using a blindfold selection technique, one district from the urban category and two districts from the rural category were selected. From these three sampled districts, the three district level hospitals were included. The names of all 21 wards in the urban districts and 55 wards in the rural districts were written in equal sized pieces of paper, using a blindfold selection technique, 2 wards from urban and 3 from rural districts were selected. From selected wards, the public health facility was included. In the selected 3 hospitals and 5 health centers, consenting HCW’s were consecutively interviewed in a private room in their working departments. The sharing of questions was limited by simultaneous interviews of eligible workers in a section (Figure 1).

### 2.4. Data Collection

A structured questionnaire was used for data collection. The questionnaire was designed using rubella information from the World Health Organization’s Rubella fact sheet and from previous researches [18,19]. The questionnaire was pre-tested by interviewing 16 HCW’s of different cadres in a health facility, the pilot data was not included in this study. Data collection was conducted by senior medical students, supervised by the principal investigator and two researchers. After obtaining written consent, the data was collected using face-to-face interviews. The following information was collected during the interviews: Socio-demographic data (age, sex, parental status and education), profession, work experience, current working department, source of information on reproductive and child health (RCH), information on awareness of the rubella vaccine, and knowledge of rubella, other infections transmitted by mother to child such as HIV and syphilis, risks, transmission routes, consequences and preventive methods. 

### 2.5. Data Analysis

The collected data was checked for completeness, coded and was entered into computer software. Statistical analysis was performed using the Statistical Package for Social Sciences (SPSS) for windows version 16 (IBM Corporation, New York, NY, USA). Awareness of rubella items was summarized in numbers and proportions of correct items among the different health care worker’s professions (Table 1). Knowledge on rubella was the main dependent variable. Adequate knowledge on rubella was defined as the correct response to five knowledge questions (Table 2). The independent variables collected were: Socio-demographics, age, sex, parental status, education, training, experience, work department, source of information for RCH. Descriptive statistics were used to summarize the data. Categorical variables were summarized by proportions (percentages). The Pearson’s chi-square or Fisher’s exact tests were used appropriately to test for differences between proportions. Univariate and multivariate logistic regression analyses were performed to obtain the dependent factors associated with HCW’s knowledge on rubella infection. Odds ratios and their 95% confidence intervals were presented to assess the strength of association between dependent and independent variables. A *p*-value of < 0.05 was considered statistically significant. For comparison, the HCW’s awareness on all infections with possible mother to child transmission (MTCT), and knowledge on HIV and syphilis was also analyzed.

### 2.6. Ethical Considerations

The study was approved by the Kilimanjaro Christian Medical University College Research Ethical Committee (CREC: Certificate number 917), an exemption for approval for the project was given by the Regional Ethics Committee (REK) in South East D Norway.

Written permission was sought from the respective District Executive Directors. Permission was also sought from the person in charge of the participating medical facilities.

Written informed consent was obtained from each participating health care worker prior to interview. Numbers were used on questionnaires to hide the participant’s identity. All information obtained was secured by the principle investigator.

## 3. Results

### 3.1. Socio-Demographic Characteristics of Study Participants

The study enrolled 126 health care workers from eight medical facilities in the three districts. The mean age of study participants was 37 years (SD = 11). The majority were nurses; 57.9 % (*n* = 73). In working experience, 75.3 % (*n* = 95) had worked more than 2 years. Less than a quarter (14.3 %) of interviewees were working in the reproductive and child health department (RCH) (Table 3). 

### 3.2. Overall Knowledge of Rubella 

The overall knowledge of rubella was 24.6 % (*n* = 31). In univariate analysis, knowledge of rubella was associated with age groups (*p* value=0.02), education level (*p* value = 0.01), experience (*p* value = 0.04) and working department (*p* value = 0.04)

In multivariate analysis, HCWs holding an advanced diploma (AOR: 7.7, 95 % CI: 1.4,41.0) or university degree (AOR: 10, 95 % CI: 2.4–42.5) were the only factors significantly associated with increased odds of rubella knowledge. Working in the out-patient department was associated with decreased odds of rubella knowledge (AOR: 0.06, 95 % CI: 0.04–0.3) (Table 3).

### 3.3. Awareness of Rubella and Other Mother to Child Transmitted Infections

The responses to rubella awareness/knowledge items by profession of HCW’s is summarized in Table 2. Among all HCWs, 62.7 % (*n* = 79) responded positively to “know or have heard about rubella infection.” Less than half gave correct responses to other rubella awareness items. The least awareness was of risk factors for rubella infection. The responses on knowledge items is summarized in Figure 2.

On awareness of all mother to child transmitted infections, HIV was mentioned by all study participants; 100 % (*n* = 126), while rubella was mentioned by 5.6 % (*n* = 7) of study participants (Table 4).

## 4. Discussion 

This study was conducted two years after the introduction of rubella vaccine (RCV) in the childhood vaccination schedule in Tanzania. To the best of our knowledge, this was the first study to assess health care worker’s knowledge on rubella infection, after RCV introduction in Tanzania. 

Rubella containing vaccine introduction was preceded by countrywide promotional campaigns, through mass media, public meetings and posters. Health care workers were involved in administering vaccines at medical facilities and in primary schools, the initial rubella vaccination coverage was above 95%. The lack of awareness on rubella is a matter of concern on the effectiveness of pre-introduction campaigns and threatens the sustenance of vaccination coverage targets [14].

The overall rubella knowledge was low, only one out of four HCW’s had adequate knowledge of rubella. This inadequacy in knowledge on rubella infection, will negatively affect the vaccination coverage in the community. As reported by Goodson and colleagues, for successful rubella elimination, high and sustained RCV coverage of greater than 95% is mandatory [20]. 

The current global rubella vaccine coverage is still low (52%). Furthermore, the vaccination coverage for other vaccines has stagnated at below 80% for the past 10 years, the success of vaccination programs depends on the leading role of HCW’s in planning for effective local vaccination programs to overcome vaccination barriers [13,21,22]. Sustaining high vaccination coverage depends on the HCW’s knowledge and attitudes on: Vaccines, their effectiveness, indications, scheduling, responding to myths and parental concerns and misinformation. Parents have little vaccine knowledge and are prone to readily available vaccine-resistant information, hence they rely on HCW’s as their trusted source of vaccine information. The better the knowledge among HCW’s, the greater the confidence in promoting vaccines [8,21]. Planning and coordinating a multi-sectorial approach to remove inequities in vaccination requires knowledgeable HCW’s. Vaccine advocacy and leadership skills are fundamental in recruiting the involvement of the community; these include children, women, men, parents, opinion, political, social-cultural, influential and religious leaders, scientists, researchers and all cadres of health care workers in implementing and sustaining vaccination targets [6,8,21,22]. Furthermore, to monitor the effectiveness of rubella vaccination, continuous surveillance on rubella and congenital rubella syndrome is a necessity. To do this, HCW’s need knowledge of the epidemiology, signs and symptoms of rubella and congenital rubella syndrome (CRS) [8,9,21]. On this basis, the Summit of Independent European Vaccination Experts (SIEVE) recommended that HCW’s vaccine knowledge, beliefs and attitudes towards vaccines should be improved for ensuring the success of vaccination programs [8]. 

In our study, age and working experience increased the odds of having knowledge on rubella infection. The young and newly employed HCW had little knowledge on rubella infection. It might be the case that the training curriculum does not cover health issues related to rubella. As reported from this study, those with better knowledge received on the job training. It is important for newly employed staff to be equipped with knowledge of the rubella infection and its prevention. A study in Thailand showed that continuous training and supportive supervision were strategies that were used to maintain high vaccine coverage [7].

The HCW’s working place increased the odds of having good knowledge of the vaccine. 

In our study, rubella knowledge was relatively high at the reproductive and child health department (RCH) (44.4%), less for in-patient (25.3%) and lowest at outpatient workers (12.1%). The HCW’s at RCH routinely perform these vaccinations, and are more likely to have more training on RCV prior to its introduction [14]. This further highlights the importance of on the job training, as HCWs at RCH and other departments have more exposure to vaccines than those in the other departments.

Other study reported the impact of training on vaccine knowledge. In a study by Brown et al. (2016) in Nigeria, a vaccine training intervention study among HCW’s improved their knowledge and practice on vaccines compared to the control groups [15]. The improvement was immediate post training, at three months and at six months evaluations. However, in evaluations at three and six months, knowledge decreased, but the scores persisted above the baseline. This study’s findings, underscores the importance of continuous training and supportive supervision in sustaining HCW’s vaccine knowledge [15]. Our study was done two years post RCV introduction. Low HCW’s knowledge was expected, if there were no follow up trainings post RCV introduction. 

Education and professional training were also factors affecting vaccine knowledge and awareness. In our study, doctors and nurses had higher knowledge and awareness than medical technologists. The basic training curricula and on the job training for HCW’s explained this difference. In the aforementioned Canadian study, which assessed the immunization education among Canadian health professional programs, it was reported that the time for immunization training varied among different health care cadres. Among the studied population, only 21% felt they had adequate teaching on vaccines during their training and were confident in discussing the vaccine with the caretakers [23]. This implied that the HCW’s knowledge on immunization depended partly on their training curriculum, but more on the experience at the workplace [23]. In our study, more than 80% of HCW’s felt that they had reliable continuous medical education at their workplace. The most frequently mentioned reliable source of information on RCH topics (including vaccination) was on the job training. 

In this study, the majority of the HCW’s were aware of other mother to child transmitted infections like HIV. These results showed that HIV received public health priority and a lot was invested to reduce the burden for HIV. To achieve this, intensive campaigns against HIV at all levels were performed including raising awareness among HCW as well as community. Regular training has ensured that all HCW’s were updated on HIV transmission and prevention. In addition, HIV and syphilis were integrated in most of the health care programs, including the reproductive and child health programs [24], which allows HCW’s to be updated on the latest information. This is an assurance that the implementation of awareness campaigns and relevant messages in continuous medical education will improve HCW’s knowledge of rubella and other vaccine-preventable infections.

## 5. Conclusions

Our study found that HCW’s knowledge and awareness on rubella was inadequate. There were significant variations in knowledge among HCW’s professions; those working in RCH were most informed on rubella infection. These findings indicate an urgent need for interventions targeted at improving HCW’s knowledge on vaccinations. This is necessary for achieving success in increasing and sustaining high vaccination coverage. 

We recommend: On the job training and supportive supervision interventions to improve vaccine knowledge among all HCW’s. Health professional training curricula need to be assessed to ensure the inclusion of immunization topics. Further studies are needed to complement the findings of this study and to provide an evidence base for policy making on future vaccine introductions.

## Figures and Tables

**Figure 1 ijerph-16-01676-f001:**
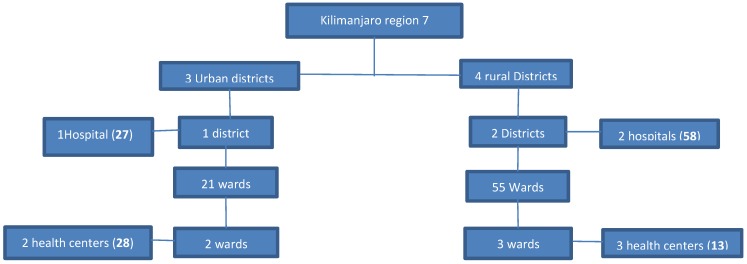
A flow chart of the sampling technique.

**Figure 2 ijerph-16-01676-f002:**
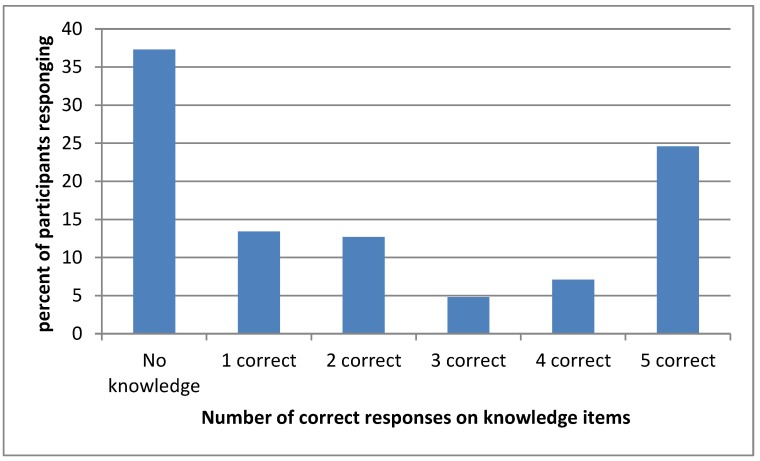
The percentage of health care workers against the number of correct responses.

**Table 1 ijerph-16-01676-t001:** Awareness of individual items on Rubella infection: Numbers and percentages of correct responses, (*n* = 79).

Items	Total (*n* %)	Doctors/Clinicians *n* (%)	Nurses *n* (%)	Technicians *n* (%)	Attendants *n* (%)
1. Have you ever heard about rubella infection?	79 (62.7)	12(60.0)	52(71.2)	9 (60.0)	6(33.3)
2. How is rubella transmitted?	46 (58.2)	8 (40.0)	34 (46.5)	2 (13.3)	2 (11.1)
3. Are there consequences of rubella infection during pregnancy?	41 (51.9)	8 (40.0)	28 (38.4)	2 (13.3)	3 (16.7)
4. Is rubella infection during pregnancy preventable/how?	52 (41.3)	5 (25.0)	39 (53.4)	4 (26.7)	4 (22.2)
5. Is rubella a cause of congenital malformations/disabilities	40 (50.6)	7 (35.0)	28 (38.3)	2 (13.3)	3 (16.7)
Which of these are risks for rubella infection during pregnancy?					
S1. Contact with a child having skin rashes?	25 (31.6)	4 (20.0)	18 (24.7)	1(6.7)	2 (11.1)
S2. Working at a child-care centre/health facility?	7(8.7)	2(10.0)	4(5.5)	0(0.0)	1(5.6)

S1 and S2: Supplementary questions.

**Table 2 ijerph-16-01676-t002:** The definition of correct responses on knowledge of rubella, *n* = 126.

Items	Correct Responses
1. Do you know/have you ever heard about rubella infection?	Yes
2. How is rubella transmitted?	Aerosol inhalation/skin contact/same as measles
3. Are there any consequences of rubella infection during pregnancy?	Yes plus correct explanation of congenital rubella syndrome
4. Is rubella infection during pregnancy preventable/how?	Yes. Immunity from prior infection/vaccination
5. Is rubella a cause of congenital malformations/disabilities	Yes, congenital rubella syndrome that affects (eye, ear, heart, brain)
Which of these are risks for rubella? Infection during pregnancy?	
S1. Contact with a child having skin rashes?	Yes
S2. Working at a child-care centre /health facility?	Yes

**Table 3 ijerph-16-01676-t003:** Background characteristics and factors associated with health care worker’s knowledge of rubella infection; *n* = 126.

Variables	*n*	Knowledge *n* (%)	*p* Value	COR (95%CI)	*p*-Value	AOR (95%CI)	*p*-Value
**Age groups (years)**							
≤30	46	5 (10.9)	0.02	1			
31–40	33	12 (36.4)		4 (1.5,15.1)	0.01	-	-
41+	47	14 (29.8)		3.5 (1.1,10.7)	0.03		
**Sex**							
Female	103	26 (25.2)		1			
Male	23	5 (21.7)	0.72	0.8 (0.3,2.4)	0.725	-	
**Self being a parent**							
Yes	94	24 (25.5)	0.68	1.2 (0.4,3.2)	0.679	-	
No	32	7 (21.9)		1			
**Education level**							
Certificate	54	8 (14.8)	<0.01	1		1	
Diploma	61	12 (23.5)		2.0 (0.8,5.2)	0.3	1.6 (0.6–4.4)	0.32
Degree	11	7 (63.6)		10.1 (2.4,42.6)			
**Medical cadre**							
Clinicians	20	5 (25.0)	0.03	1		1	
Nurses	73	24 (32.9)		1.5 (0.5–4.5)	0.5	2.7 (0.7–11.5	0.16
Medical attendants	18	2 (11.1)		0.4 (0.06–2.2)	0.28	0.6 (0.1–5.2)	0.64
**Experience(years)**							
0-2	31	2 (6.5)	0.01	1			
3–9	30	6 (20.0)		3.6 (0.7,19.6)	0.135	-	
10+	65			7.9 (1.7,36.3)	0.008		
**Work station**							
RCH (hospital)	18	8 (44.4)	0.04	1		1	
Outpatient	33	4 (12.1)		0.17 (0.0,0.7)	0.014	0.04 (0.1–0.3)	0.01
In patient (wards)	75	19 (25.3)		0.42 (0.2,1.2)	0.12	0.3 (0.1–1.0)	0.05
**RCH information** **Source–reading**							
Yes	46	16 (34.8)	0.04	1			
No	80	15 (18.8)		0.4 (0.2,1.0)	0.047	-	
**Shares phone contact** **with clients**							
Yes	101	29 (28.7)	0.04	1		1	
No	25	2 (8.0)		0.22 (0.1–1.0)	0.046	0.5 (0.0–1.0)	0.05

**Table 4 ijerph-16-01676-t004:** Awareness of all mother to child transmissible infections responses (*n* = 126).

Type of Infection	Response Frequency (*n*)	Response Rate *n* (%)	Doctors *n* (%)	Nurse *n* (%)	Technicians *n* (%)	Medical Attendants *n* (%)
HIV	126	100	20(100)	73(100)	15(100)	18(100)
Syphilis	109	86.5	17(85.0)	68(93.2)	9(60.0)	16(88.9)
Gonorrhea	60	47.6	8(20.0)	35(47.9)	6(40.0)	12(66.7)
HBV	18	14.3	3(15.0)	11(15.1)	2(13.3)	2(11.1)
Rubella	7	5.6	1(5.0)	5(6.8)	0(0.0)	1(5.6)
Toxoplasmosis	5	4	2(10.0)	0(0.0)	1(5.6)	2(11.1)
CMV	1	0.8	0(0.0)	0(0.0)	0(0.0)	1(5.6)

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
