# Peer review of "Awareness and Factors Associated with Health Care Worker’s Knowledge on Rubella Infection: A Study after the Introduction of Rubella Vaccine in Tanzania"

_ijerph, 2019, doi:10.3390/ijerph16101676_

Round 1

Reviewer 1 Report

Thank you for giving me this opportunity to read this interesting study. I hope my suggestions would help you improve this manuscript.

Abstract

Some sections are missing.

Introduction

More details are needed about the rubella infection or vaccination status in Tanzania.

Stronger rationale for the needs of this study, that is assessing health care worker’s knowledge on rubella infection, is necessary.

Methods

In the study design and area section, why did you chose the three regions and are there any differences of these regisions from each other? Are they representative? More details about these regions are necessary.

In the abstract and in the “study population” section, it seems like this study used all health care workers. This may need to be revised, as this study first selected a few facilities.

In the data collection, you may clarify if it was a qualitative interview or just conducting quantitative surveys via interviews.

Also, citations from which the interview questions were drawn are needed. In the methods, no citations are employed. For example, where did you get the knowledge-related items?

Results

Table 3 would be better to read if it is revised in some ways.

Results are repetitively presented in the tables and texts. Redundant information needs to be removed.

Some typos or grammatical errors need to be fixed.

Author Response

Thank you for the commends,

I have uploaded the word file for reviewer number 1 responses.

Reviewer 2 Report

The title of the research is concise and informative and indicates the research approach used.

Abstract provides detailed methodological information.

In the background, the authors should clearly justify why they wanted to investigate the health care workers knowledge on rubella vaccine

On line 82, the authors mentioned all health care workers. What do they mean by all health care worker? Who are included? They needed to be very specific here.

A flow chart presenting the sampling technique would have been useful here. How they determine the sample size. There is no information how many health care workers working in those hospitals. Did they manage to include everyone from those hospitals? Please provide detailed information.

There is no information regarding the validation of the questionnaire. How the questionnaire was validated?

Result section is well written and presented

The discussion section lacks balanced argument and no evidence of critical evaluation of similar research findings. 

The implications of this research findings are not clear.They needed to be written in a clear format.

Author Response

Thank you for the comments,

I have uploaded a word file with reviewer 2 responses.

Round 2

Reviewer 2 Report

Thanks for resubmitting the work. It looks much better now.

Author Response

Thank you for the comment.

I have noted and corrected reference number 5, as highlighted in the reference section.